# Systematic Cardiovascular Screening in Olympic Athletes before and after SARS-CoV-2 Infection

**DOI:** 10.3390/jcm11123499

**Published:** 2022-06-17

**Authors:** Viviana Maestrini, Domenico Filomena, Lucia Ilaria Birtolo, Andrea Serdoz, Roberto Fiore, Mario Tatangelo, Erika Lemme, Maria Rosaria Squeo, Ruggiero Mango, Giuseppe Di Gioia, Francesco Fedele, Gianfranco Gualdi, Antonio Spataro, Antonio Pelliccia, Barbara Di Giacinto

**Affiliations:** 1Department of Clinical, Internal, Anesthesiologic and Cardiovascular Sciences, Sapienza University of Rome, 00161 Rome, Italy; domenico.filomena@uniroma1.it (D.F.); ilariabirtolo@gmail.com (L.I.B.); francesco.fedele@uniroma1.it (F.F.); 2Institute of Sport Medicine, Sport and Health, National Italian Olympic Committee, 00197 Rome, Italy; andreaserdoz@gmail.com (A.S.); roberto.fiore@mail.com (R.F.); tatangelo.mario@libero.it (M.T.); erikalemme@msn.com (E.L.); marysqueo@yahoo.it (M.R.S.); ruggiero.mango@gmail.com (R.M.); dottgiuseppedigioia@gmail.com (G.D.G.); gianfranco.gualdi@uniroma1.it (G.G.); antonio.spataro@sportesalute.eu (A.S.); ant.pelliccia@gmail.com (A.P.); ext_barbara.digiacinto@sportesalute.eu (B.D.G.); 3Department of Biomedicine, Neurosciences and Advances Diagnostics, University of Palermo, 90133 Palermo, Italy

**Keywords:** COVID-19, SARS-CoV-2, athletes, cardiovascular evaluation, CMR, return-to-play (RTP)

## Abstract

Conflicting results on the cardiovascular involvement after SARS-CoV-2 infection generated concerns on the safety of return-to-play (RTP) in athletes. The aim of this study was to evaluate the prevalence of cardiac involvement after COVID-19 in Olympic athletes, who had previously been screened in our pre-participation program. Since November 2020, all consecutive Olympic athletes presented to our Institute after COVID-19 prior to RTP were enrolled. The protocol was dictated by the Italian governing bodies and comprised: 12-lead ECG, blood test, cardiopulmonary exercise test (CPET), 24-h ECG monitoring, and spirometry. Cardiovascular Magnetic Resonance (CMR) was also performed. All Athletes were previously screened in our Institute as part of their periodical pre-participation evaluation. Forty-seven Italian Olympic athletes were enrolled: 83% asymptomatic, 13% mildly asymptomatic, and 4% had pneumonia. Uncommon premature ventricular contractions (PVCs) were found in 13% athletes; however, only 6% (*n* = 3) were newly detected. All newly diagnosed uncommon PVCs were detected by CPET. One of these three athletes had evidence for acute myocarditis by CMR, along with Troponin raise; another had pericardial effusion. No one of the remaining athletes had abnormalities detected by CMR. Cardiac abnormalities in Olympic athletes screened after COVID-19 resolution were detected in a minority, and were associated with new ventricular arrhythmias. Only one had evidence for acute myocarditis (in the presence of symptoms and elevated biomarkers). Our data support the efficacy of the clinical assessment including exercise-ECG to raise suspicion for cardiovascular abnormalities after COVID-19. Instead, the routine use of CMR as a screening tool appears unjustified.

## 1. Introduction

Cardiovascular involvement after Severe Acute Respiratory Syndrome CoronaVirus-2 (SARS-CoV-2) infection in patients without symptoms or mildly symptomatic generated concerns on the safety of the return-to-play (RTP) in the athletic population, related to the risk associated with an underlying, silent cardiac injury. Preliminary data provide an alarmingly high prevalence of cardiovascular involvement in asymptomatic athletes also [1,2], whereas subsequent studies on a wider cohort showed a low prevalence of cardiac involvement in those athletes, with a benign course [3,4,5,6,7,8,9,10,11]. Of note, all the studies published so far have a cross-sectional design and are mainly based on imaging findings. Thus, uncertainties about the real prevalence of cardiac complications after SARS-CoV-2 infection still exist, fueling the ongoing debate on the optimal diagnostic protocol to evaluate competitive athletes recovering from the COVID-19.

Different algorithms suggesting the modalities of the CV screening before the return-to-play (RTP) have been released by scientific societies and sport governing organizations [12,13,14,15]. However, these recommendations are largely based on wise clinical judgment and expert opinion. Moreover, the implementation of the proper diagnostic pathway to advise a safe RTP is of pivotal relevance in professional and Olympic athletes, in whom cardiac integrity is the prerequisite for maintaining the excellence in performance, and also achieving the economic, social, and media-related benefits. 

In order to provide solid data in this field, we sought to assess the prevalence of CV abnormalities in Olympic athletes, previously screened within our Olympic medical program and re-evaluated immediately after COVID-19 resolution, according to the comprehensive protocol dictated by the Italian Ministry of Health [14], with the addition of CMR.

## 2. Methods

### 2.1. Study Population

All Olympic athletes who were diagnosed with SARS-CoV-2 infection were consecutively evaluated at the Institute of Sport Medicine prior to resuming their training, starting from November 2020. All athletes were evaluated in the pre-vaccination period. The diagnosis of COVID-19 was made by a positive oro/nasopharyngeal throat swab (NPS) for SARS-CoV-2 by reverse-transcriptase-polymerase chain reaction. 

The protocol of the CV evaluation was performed in agreement with the Italian Ministerial Decree released after the first wave of pandemic [14]. The protocol for competitive athletes included: -Physical evaluation and medical history: specifically, the duration of infection (time from the first positive NPS to the first negative), the time from the negative NPS to evaluation, the presence of symptoms related to the infection [16], and eventually any medications were collected.-Blood tests: the list of the blood tests is listed in the Appendix A.-Pulmonary function tests (PFTs): parameters measured are listed in the Appendix A.-12-lead resting electrocardiogram (ECG): ECG patterns were analyzed according to the international criteria [17]. In case of abnormalities, ECGs post-COVID-19 were compared to the ones recorded previously.-Transthoracic echocardiogram (TTE): left and right ventricle (LV and RV) dimension, wall thickness, global and regional systolic function, indexes of diastolic function, as well as the presence of pericardial effusion were evaluated. Imaging interpretation was based off the international recommendation [18,19].-Cardiopulmonary exercise test (CPET): cardiopulmonary performance parameters were collected, and are listed in the Appendix A [20]. Exercise-induced ventricular arrhythmias were evaluated as well.-24-h ECG monitoring: the occurrence of rhythm and conduction abnormalities, and supra-ventricular and ventricular arrhythmias were investigated. The burden of premature ventricular contractions (PVCs) was arbitrarily classified as <50, 50–500, >500 PVCs/24 h.

Ventricular arrhythmias were classified as common or uncommon pattern, as previously suggested [21]. Newly diagnosed PVCs were those detected for the first time after COVID-19 (not present in previous evaluation). 

In addition to the tests included in the recommended protocol, a contrast CMR was performed. The CMR protocol, including mapping sequences, is described in detail in the Appendix A [22,23]. CMR images were analyzed off-line, blinded to clinical data. The following parameters were collected: indexed LV and RV end-diastolic (EDV) and end-systolic volumes (ESV), LV mass and ejection fraction (LVEF), native myocardial and blood T1 mapping values, extracellular volume (ECV), and T2 mapping. The late gadolinium enhancement (LGE) presence and pattern were also evaluated. The CMR diagnosis of myocarditis was made in accordance with expert recommendations of CMR in non-ischemic myocardial inflammation [24]. 

All cardiovascular exams were performed on the same day at the Institute of Sport Medicine.

Notably, all athletes had previously been evaluated at our Institute, as part of our pre-participation screening prior to the Olympic Games, in the year before the most recent post-COVID-19 re-evaluation. A subgroup of these athletes had also previously undergone non-contrast CMR, prior to SARS-CoV-2 infection, as part of a different research project. 

The study design of the present investigation was evaluated and approved by the Review Board of the Institute, and conforms to the ethical guidelines of the 1975 Declaration of Helsinki. All athletes included in this study were fully informed of the types and nature of the evaluation and signed the consent form, pursuant to Italian law and the Institute policy. 

All clinical data assembled from the study population are maintained in an institutional database. 

### 2.2. Statistical Analysis

Dichotomous variables were expressed as simple frequencies (*n*) and percentages (%), and continuous variables were summarized as mean and standard deviation (SD) or median and 25th–75th percentiles, when appropriate. Normal distribution was tested by the Shapiro–Wilk test. Statistical tests were two-sided, and *p* < 0.05 was considered significant. Differences in CMR parameters before and after COVID-19 were evaluated by t-test for paired samples. Statistical analysis was performed using SPSS (IBM SPSS Statistics for Windows, ver.27, IBM, Armonk, NY, USA).

## 3. Results

Forty-seven Olympic athletes were enrolled in the RTP program. Demographic data are shown in Table 1. The mean age was 26 ± 4 years, and 32 (68%) were male. Athletes were largely engaged in endurance disciplines (53%), or power (28%), and less in mixed (19%). 

In our cohort, the median duration of the infection was 14 (11–22) days, and the time between the first negative NPS and the RTP evaluation was 9 (6–13) days. The vast majority of athletes during COVID-19 were only mildly symptomatic (83%), and 13% were completely asymptomatic. Only a small subset (*n* = 2, 4%) had clinically evident pneumonia, which was treated with antibiotics and corticosteroids without the need of hospitalization. One athlete with pneumonia also experienced deep vein thrombosis, treated with low-molecular-weight heparin. 

Symptoms during the infection (Table 1) mostly comprised fever, ageusia/anosmia, myalgia, headache, and marked asthenia. At the time of the RTP evaluation, symptoms were reported by one male athlete with previous pneumonia, complaining of persistent fatigability, and two female athletes reporting palpitations. One of them, an open-water swimmer, had a significantly increased troponin value. No other blood test abnormalities were detected, including white blood count, IL-6, and C-reactive protein (CPR). 

Spirometry was negative in all subjects, except for a 26-year-old swimmer with known allergic asthma who had a modest obstructive pattern, already present a year before the infection, reversible after bronchodilator inhalation.

Comparing the ECG acquired pre- and post-COVID-19, no new electrocardiographic abnormalities were revealed (Table 2). ECG abnormalities present at pre-COVID-19 evaluation comprised two athletes (4%) with anterior T-wave inversion (in V2-V3), and three athletes (6%) with isolated left axis deviation. All of them had otherwise normal cardiac evaluation (negative family history, no symptoms, normal imaging test, no arrhythmias). 

The CPET showed a normal cardiopulmonary response in all athletes (Table 2), without ST-T abnormalities. The mean heart rate, systolic and diastolic blood pressure, and VO2 max were, respectively, 169 ± 9 bpm, 174 ± 17 mmHg, 75 ± 8 mmHg, and 42 ± 6 L/Kg/min. Of note, *n* = 6 (13%) athletes presented exercise-induced PVCs during the CPET. Comparing the results of the exercise test before and after COVID-19, we found that in three athletes, PVCs were already present at the evaluation before COVID-19. On both occasions (pre- and post-COVID-19), PVCs were rare and isolated, and without associated structural heart disease, symptoms, or family history. Instead, newly diagnosed PVCs were observed in three (6%) athletes, all with an uncommon pattern. 

Twenty-four-hour ECG monitoring revealed isolated supraventricular extrasystoles in 40 (85%) athletes, of which, 93% were less than 500. PVCs were detected in 32 athletes (68%), of which, 81% were less than 50 isolated PVCs (Table 2). Only three athletes had complex ventricular arrhythmias (also present at CPET).

Echocardiogram revealed one female athlete with mild pericardial effusion, absent in the pre-COVID-19 evaluation. None of the athletes showed impaired LV systolic or diastolic dysfunction post-COVID-19. 

CMR confirmed that LV and RV cavity dimensions and function were within the normal limits in all of the athletes (in relation to body surface area, sex, and type of sport) (Table 3). Tissue characterization revealed an acute myocarditis (in one athlete, 2%), associated with a raise in troponin. After contrast administration, no one had pathological LGE. Four athletes (8.5%) had non-specific LGE at the inferior RV insertion point. 

A comparative assessment of CMR findings available in a subgroup of 18 athletes (prior and after COVID-19) did not show any significant difference in terms of LV and RV dimensions, function, and native T1 and T2 mapping (Table 4). 

Summarizing, in the overall cohort of 47 athletes, only 3 (6%) athletes presented new cardiac abnormalities after COVID-19 (Figure 1). An open-water female swimmer presented, after an asymptomatic SARS-CoV-2 infection, with rare palpitations and frequent exercise-induced PVCs (with left bundle branch block morphology and inferior axis), including two short (three beats) bursts of non-sustained ventricular tachycardia (NSVT) during exercise. ECG revealed anterior T-wave inversion (already present before COVID-19). High-sensitivity troponin T was increased (75 pg/mL, cut-off < 14 pg/mL). The echocardiogram was unremarkable. Contrast CMR provided evidence for acute myocarditis, based on increased native T1 and T2 values at the mid-inferior wall and inferior septum (Figure 1A1–A5). The athletes had a non-contrast CMR prior the SARS-CoV-2 infection, and the mapping was within normal limits.

A female fencer experienced a mildly symptomatic COVID-19 with fever, headache, and myalgia. During the RTP evaluation, she complained of palpitation, and presented frequent exercise-induced polymorphic PVCs, isolated and in couplets. Echocardiogram and CMR revealed a new mild pericardial effusion, without signs of pericardial and myocardial inflammation (Figure 1B1–B5). Both athletes entered a close follow-up (re-evaluation at 3 months), and were temporarily withdrawn from competitions. 

Finally, a soccer male player presented, after mildly symptomatic COVID-19 (fever), with frequent exercise-induced PVCs, often as R-on-T, and couplets. The 24-h ECG monitoring showed a modest arrhythmic burden (total PVCs < 500, but mostly exercise-induced), often as R-on-T. The 12-lead ECG showed T-wave inversion in V2–V3 (already present before COVID-19). Blood test, TTE, and CMR were normal. The athlete was advised to temporarily withdraw from competition, and entered a close follow-up program (Figure 1C1–C5).

After three months, the three athletes were re-evaluated. The long-distance swimmer was asymptomatic, and the troponin and the CMR were negative. However, the athletes still had complex ventricular arrhythmias (NSVT) induced by exercise. Thus, they were advised to prolong the recovery period before resuming competition. The other two athletes were asymptomatic, but both still had uncommon ventricular arrhythmias. Thus, both of them remained on follow-up.

## 4. Discussion

Conflicting results have been reported over the last months regarding the prevalence of CMR findings (from 0.4 up to 15%) suggestive for myocardial involvement in competitive athletes during the recovery after even an asymptomatic or mildly symptomatic SARS-CoV-2 infection [1,3,4,5,6,7,8,9,10]. A recent metanalysis estimated a pooled prevalence of COVID-19-related myocarditis among athletes, ranging from 1 to 4% [25]. However, given the cross-sectional nature of the previous study designs, with different times of evaluation and protocol applied, and different criteria for recruiting athlete cohorts, uncertainty still exists regarding how to advise a safe return-to-play in competitive athletes. In addition, in some cases, the cardiac abnormalities reported were based on isolated abnormal findings at imaging testing. 

Our evaluation was based on the first mandatory protocol dictated for competitive athletes aiming for a return-to-play after COVID-19 by the Italian sport governing bodies. In this study, CMR was performed in addition to this protocol, due to accumulating reports describing a large, even clinically unsuspected, prevalence of myocardial injury induced by COVID-19. The strength of this study is the fact that all athletes were recruited consecutively (without any selection bias) and evaluated in our Institute at least once prior to infection. Thus, we could compare the results of the post-COVID-19 evaluation with the previous medical records. In addition, a comparison of CMR scans performed pre- and post-COVID-19 was also possible in a subgroup of athletes.

Based on this unique dataset, we found that new ventricular arrhythmias were detected in 6% of athletes. Specifically, all three athletes presented a new onset of ventricular arrhythmias with an uncommon pattern, of which, one was eventually diagnosed as acute myocarditis. PVCs were detected by exercise ECG, and the burden was evaluated by 24-h Holter monitoring. Eventually, in one athlete, the diagnosis of acute myocarditis was confirmed by CMR and an increase in troponin. CPET was performed before obtaining troponin and CMR results, since these tests were performed for screening evaluation. In all other cases, tissue characterization by CMR was unremarkable. 

Therefore, our data are in line with those suggesting that CMR has an increased diagnostic yield only when performed upon clinical indication. Furthermore, we reassure the community of professional athletes and their sport clinicians regarding the efficacy of the clinical protocol implemented for the RTP, because we did not find any abnormal CMR findings in otherwise asymptomatic athletes with negative clinical evaluation. 

Our data diverge from certain previous studies, most likely because of the inclusion criteria adopted and/or the protocol applied and/or the definition of cardiac involvement. Rajpal et al. [1] reported a study on a limited cohort of 26 competitive athletes. CMR was performed in all subjects on top of ECG, echocardiogram, and troponin. The authors reported an unexpected prevalence of myocarditis (15%, the highest percentage described so far), based on increased T2 mapping values (with otherwise normal T1 mapping) and LGE, with only two showing symptoms. 

Another small retrospective study on 59 competitive athletes after COVID-19 found only 3% of subjects with myocarditis based on CMR findings [7]. One of the athletes with myocarditis subsequently developed LV dysfunction. 

Two retrospective studies including larger cohorts of athletes reported a lower prevalence of cardiac injuries. Starekova et al. found a prevalence of myocarditis of 1.4% (*n* = 2) on a group of 145 competitive athletes who had undergone a blood test, ECG, echocardiogram, and CMR [6]. One of these athletes had symptoms and a raise in troponin, with extensive acute myocardial damage at CMR. The other one had minor tissue abnormalities. The other study retrospectively analyzed 789 professional athletes [3]. According to the protocol advised, CMR and stress echocardiography were requested upon clinical indication, based on abnormal ECG, or echocardiogram, or increased troponin. Only 30 subjects (3.8%) had an indication to undergo further examinations, and myocarditis was detected in 0.6% and pericarditis in 0.25% of the overall athletic cohort. 

Another three studies found no case of acute myocarditis. Malek et al. examined 26 athletes and found only minor changes (isolated small increase in T2 mapping) in 5, and the criteria for acute myocarditis were not fulfilled in anyone [6]. The study population reported by Vago et al. [7] was limited to 13 subjects, and none of them showed any cardiac involvement by CMR. Similarly, Hendrickson et al. performed a RTP evaluation of 137 collegiate athletes. Five subjects (of which, four presented with increased cardiac troponins) underwent CMR, which did not detect myocardial injury or myocarditis [10]. Moulson et al. recruited 3018 collegiate athletes recovering from COVID-19. Among all subjects, 317 underwent CMR, of which, 198 were unselected, whereas 119 underwent CMR based on clinical indication. Acute myocarditis was detected in 2% (0.2% of the overall cohort) [8]. The authors also found that the diagnostic yield of CMR for cardiac involvement related to COVID-19 was 4.2 times higher in cases of a clinically indicated CMR, versus a primary screening CMR. 

Daniels et al. reported the results of an extensive multicentric CMR screening after COVID-19 in athletes [9]. The authors recruited 1597 competitive collegiate athletes, and compared different diagnostic algorithms. The strategy of using CMR imaging for screening in all athletes regardless of cardiac symptoms or other cardiac testing results increased the prevalence to 2.3%, a 7.4-fold increase from the symptom-driven strategy, and a 2.8-fold increase over the ECG, echocardiogram, and troponin strategy. However, the clinical meaning of CMR tissue changes out of a clear clinical presentation is currently not known. In all the cited studies, a few sporadic cases of pericardial involvement were observed, whereas only Brito et al. [2] reported an unusually (one of three) high prevalence of pericarditis based on CMR. In that study, no myocarditis was detected. All the aforementioned studies were focused mostly on imaging data, with the latest ones having the aim to solve the problem of the diagnostic role of CMR in this setting. A more recent study compared a group of 147 athletic individuals after SARS-CoV-2 with a control group of athletes and less active control. The prevalence of myocarditis was 1.4%, and no difference was observed between the three groups in terms of mapping [11].

In all the previous studies, CPET and arrhythmia evaluation (both with Holter and exercise ECG) were not performed nor reported. According to the Italian protocol, instead, 24-h ECG monitoring and exercise tests are integral parts of the post-COVID-19 evaluation for competitive athletes. Thus, CPET was important to confirm the normal cardiac performance in all subjects examined, and was able to identify new ventricular arrhythmias in a subset of them, not present in the pre-COVID-19 evaluation, which were predictive in few cases for an underlying cardiac condition requiring temporarily withdrawal from competitions. On the other and, the approach based on ECG, echocardiogram, and troponin may fail in the identification of a subset of patients with arrhythmic complications, who may require further investigation as CMR.

Data on cardiovascular evaluation including exercise stress tests in adults are scarce. Gervasi et al. reported data on spirometry, ECG, stress-test ECG with oxygen saturation monitoring, and echocardiogram in 30 athletes, and did not reveal any relevant anomalies [26]. Cavigli et al. found 3.3% cardiac complication (one myo-pericarditis and two pericarditis) after COVID-19 in 90 competitive athletes. Specifically, CPET demonstrated uncommon arrhythmias in only one athlete (1%) [27]. Both studies were cross-sectional in design. A very low prevalence of uncommon ventricular arrhythmias was also confirmed in the pediatric population recovering from SARS-CoV-2 infection [28].

### Limitations

Our investigation has several limitations to be elicited. The main limitation is inherent to our study population, comprising a relatively limited number of elite competitive athletes in preparation for the Olympic Games. The low number of athletes enrolled represents a major limitation, and precludes the possibility to evaluate the diagnostic yield of both CMR and exercise tests in a large cohort of athletes.

The majority of subjects were asymptomatic or mildly asymptomatic. Accordingly, this cohort is not representative of those subjects with a more severe COVID-19 course. Finally, not all the athletes underwent CMR before COVID-19, and this was, in any case, performed without contrast. CPET was not available in the evaluations performed prior to COVID-19 because athletes are usually investigated by exercise stress test. Thus, a comparison of before and after COVID-19 for CPET was not possible. 

## 5. Conclusions

Among the group of 47 Italian Olympic athletes recovering from SARS-CoV-2 infection and undergoing a comprehensive CV evaluation, only a minority (6%) had newly detected cardiac abnormalities requiring a temporary withdrawal from competition. Specifically, all three athletes presented a new onset of ventricular arrhythmias with uncommon patterns detected by CPET, and only one (2%) had evidence of acute myocarditis at CMR. Despite the limited number of athletes precluding the evaluation of the diagnostic yield of CMR and exercise tests, this unique dataset demonstrated that cardiac complication could occur after COVID-19 with a low prevalence, and the routine use of CMR as a screening tool appears unjustified.

## Figures and Tables

**Figure 1 jcm-11-03499-f001:**
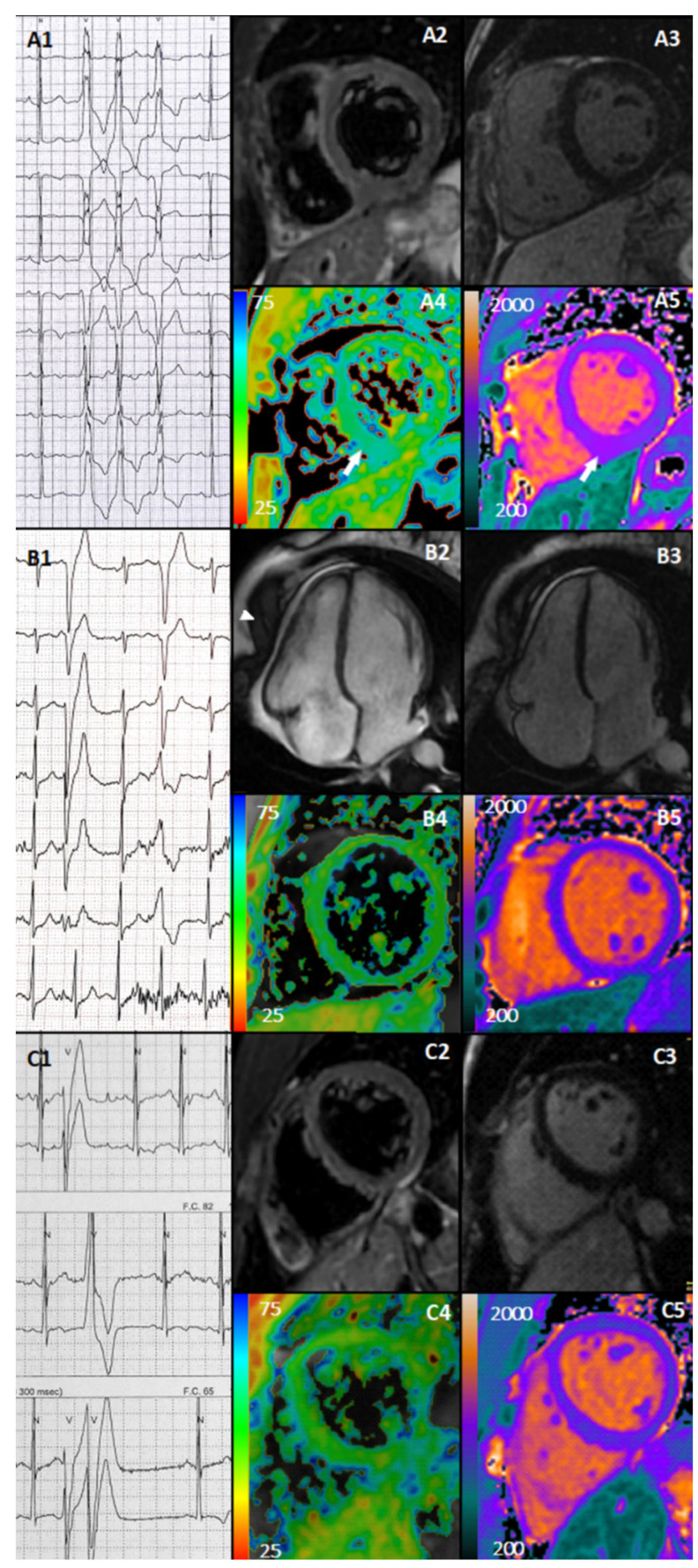
Newly detected abnormal findings. Case 1. Athletes presented with exercise-induced NSVT with LBBB morphology, inferior axis and transition in V3 (**A1**), negative T2-weighted (**A2**), absence of LGE (**A3**), increased T2 mapping (**A4**), and native myocardial T1 mapping (**A5**) at the mid-inferior wall and inferior septum (white arrows). Case 2. Athlete presented with exercise-induced polymorphic PVCs with LBBB morphology with transition in V6 (first one) and in V4 (second one) (**B1**), cine CMR with mild pericardial effusion (arrows) (**B**2), negative LGE (**B3**), normal T2 mapping (**B4**), and T1 myocardial native mapping (**B5**). Case 3. Athlete presented with exercise-induced polymorphic PVCs, often R on T, and a couplet (**C1**) and negative T2-weigthed sequence (**C2**), negative LGE (**C3**), normal T2 mapping (**C4**), and native myocardial T1 mapping (**C5**).

**Table 1 jcm-11-03499-t001:** Study population and clinical presentation of SARS-CoV-2 infection.

Parameter	Athletes (*n* = 47)
Age, y.o.	26 ± 4
Male sex, *n* (%)	32 (68)
Caucasian, *n* (%)Afro-Caribbean, *n* (%)	45 (96)2 (4)
Years of training, y	16 (10–20) *
Hours of training/week, h/week	22 (18–26) *
Weight, kg	78 ± 15
Height, cm	180 ± 11
BSA, m^2^	1.9 ± 0.23
BMI, kg/m^2^	24 ± 3
SBP, mmHg	112 ± 11
DBP, mmHg	71 ± 9
HR, bpm	57 ± 12
Symptoms, *n* (%)	41 (87)
Ageusia, *n* (%)	19 (40)
Anosmia, *n* (%)	15 (32)
Fever, *n* (%)	17 (36)
Dyspnea, *n* (%)	3 (6)
Diarrhea, *n* (%)	3 (6)
Chest pain, *n* (%)	1 (2)
Faintness, *n* (%)	12 (26)
Headache, *n* (%)	20 (43)
Myalgia, *n* (%)	13 (28)
Cold, *n* (%)	12 (26)
Palpitations, *n* (%)	2 (4)

* Median (25th–75th percentiles), BMI: body mass index; BSA: body surface area; DBP: diastolic blood pressure; HR: heart rate; SBP: systolic blood pressure.

**Table 2 jcm-11-03499-t002:** Screening results: ECG, 24-h Holter monitoring, and CPET.

Parameter		Athletes (*n* = 47)
12-lead ECG		
TWI, *n* (%)		2 (4)
LAD, *n* (%)		3 (6)
Newly detected abnormal ECG, *n* (%)		0 (0)
24-h ECG Holter monitoring (12 lead)		
SVPCs, *n* (%)		40 (85)
SVPCs >500/h, *n* (%)		3 (6)
PVCs, *n* (%)		32 (68)
PVCs	<50/24 h, *n* (%)	26 (55)
	50–500/24 h, *n* (%)	2 (4)
	>500/24 h, *n* (%)	4 (9)
Cardiopulmonary test (CPET)		
HR max, bpm		169 ± 9
SBP max, mmHg		174 ± 17
DBP max, mmHg		75 ± 8
Watt max, W		304 ± 94
VO_2_ max, L/min		3223 ± 781
VO_2_/Kg max, mL/kg/min		42 ± 6
VO_2_/Kg AT, mL/kg/min		22 ± 4
VE max, L/min		111 ± 34
VE/VCO_2_ slope		25 ± 5
VO_2_/HR, L/min/bpm		19 ± 5
RER max		1.2 ± 0.1
Exercise-induced VA, *n* (%)		6 (13)
Complex exercise-induced VA, *n* (%)		3 (6)
Newly diagnosed exercise-induced VA, *n* (%)		3 (6)

AT: anaerobic threshold; CVA: complex ventricular arrhythmias; DPB max: maximal diastolic blood pressure; ECG: electrocardiogram; HR max: maximal heart rate; LAD: left axial deviation; PVCs: premature ventricular contractions; RER: respiratory exchange ratio; SBP max: maximal systolic blood pressure; SVPCs: supra-ventricular premature contractions; TWI: T wave inversion; VA: ventricular arrhythmias; VE max: maximal ventilation; VE/VCO_2_ slope: ventilatory efficiency slope; VO_2_ max: maximal oxygen uptake.

**Table 3 jcm-11-03499-t003:** Morphologic and functional cardiac findings.

Parameter	Athletes (*n* = 47)
Cardiac magnetic resonance	
LV EDVi, mL/m^2^	111 ± 18
LV ESVi, mL/m^2^	49 ± 10
LV SVi, mL/m^2^	62 ± 10
LV EF, %	56 ± 4
RV EDVi, mL/m^2^	107 ± 17
RV ESVi, mL/m^2^	46 ± 10
RV SVi, mL/m^2^	61 ± 10
RV EF, %	57 ± 4
LAAi, cm^2^/m^2^	13 ± 2
RAAi, cm^2^/m^2^	13 ± 2
IVST, mm	9 ± 1.2
PWT, mm	8 ± 1.2
Pathological LGE, *n* (%)	0 (0)
RV insertion point LGE, *n* (%)	4 (8.5)
Pericardial effusion, *n* (%)	1 (2)
Parametric Mapping	
T1 Blood, ms	1499 ± 105
T1 Myo, ms	950 ± 36
T2 Myo, ms	50 ± 2
ECV, %	0.27 ± 0.02
Positive Lake Louise Criteria, *n* (%)	1 (2)
Echocardiography	
E/A	1.7 ± 0.5
E, cm/s	77 ± 16
A, cm/s	46 ± 10
e’, cm/s	11 ± 3
E/e’	10 ± 3
TAPSE, mm	26 ± 4
sPAP, mmHg	25 ± 3

ECV: extracellular volume; EDVi: end diastolic volume index; ESVi: end systolic volume index; EF: left ventricular volume index; LAAi: left atrium area index; LGE: late gadolinium enhancement; LV: left ventricle; LV mass/i: left ventricular mass index; myo: myocardium; LWT: lateral wall thickness; IVST: interventricular septum thickness; PWT: posterior wall thickness; RAAi: right atrium area index; RV: right ventricle; SVi: stroke volume index; sPAP: systolic pulmonary arterial pressure; TAPSE: tricuspid annular plane systolic excursion.

**Table 4 jcm-11-03499-t004:** CMR finding before and after SARS-CoV-2 infection.

Parameter	Before SARS-CoV-2 *n* = 18	After SARS-CoV-2 *n* = 18	*p*
LV EDVi, mL/m^2^	118 ± 19	119 ± 19	0.493
LV ESVi, mL/m^2^	51 ± 13	53 ± 11	0.069
LV SVi, mL/m^2^	67 ± 9	66 ± 9	0.516
LV EF, %	57 ± 5	56 ± 4	0.081
RV EDVi, mL/m^2^	117 ± 17	116 ± 17	0.494
RV ESVi, mL/m^2^	55 ± 21	51 ± 10	0.333
RV SVi, mL/m^2^	66 ± 9	65 ± 10	0.648
RV EF, (%)	56 ± 4	56 ± 4	0.957
T1 Mapping Blood, ms	1438 ± 73	1466 ± 98	0.26
T1 Mapping Myo, ms	935 ± 29	939 ± 48	0.746
T2 Mapping Myo, ms	51 ± 3	48 ± 2	0.099
RV insertion point LGE, *n* (%)	NA	2 (11)	NA

EDVi: end diastolic volume index; ESVi: end systolic volume index; EF: ejection fraction; LGE: late gadolinium enhancement, LV: left ventricle; LV mass/i: left ventricular mass index; myo: myocardium; NA: not applicable; RV: right ventricle; SVi: stroke volume index.

## Data Availability

The data presented in this study are available on request from the corresponding author.

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
