# Peer review of "Systematic Cardiovascular Screening in Olympic Athletes before and after SARS-CoV-2 Infection"

_jcm, 2022, doi:10.3390/jcm11123499_

Round 1

Reviewer 1 Report

In the present study, the authors reported the results from a systematic cardiovascular screening performed in athletes who recovered form SarS-CoV2 infection. This is timely and interesting study, addressing the concerns raised on myocardial involvement due to COVID-19 and the subsequent potential consequences on professional athletes. The study design is appropriate, all patients underwent a comprehensive evaluation; of note, all of them recieved a similar evaluation (albeit with CMR just in a subgroup) prior to SarS-CoV2 infection. The manuscript is well written and results clearly reported. Out of 47 patients, only 2 were found having abnormal CMR findings (pleural effusion in one case, myocarditis in the other); of note, both patients had newly detected abnormality in other non-imaging test. The authors concluded highlighting the importance of of aCV evaluation priori to RTP in athletes after  SarS-CoV2 infection, albeit suggesting a role for CMR just as a second-line, non-screening, test. I have just few minor comments:

- I cannot see the cited "supplementary data" accompanying the manuscript

- I might be worth adding few details: were all patients evaluated in the pre-vaccination periood? Was there any peculiar course of SarS-CoV2 infection in the 3 patients with newly detected CV abnormalities?

- Few typos (i.e. ap-pears, map-ping etc.)

Author Response

We would like to thank the Reviewers for the evaluation of our manuscript. We are pleased to read that the Reviewers have considered our paper of potential interest upon satisfactory responses to the comments and issues raised during the review process. We revised the manuscript which now includes additional data. Please, find our point-by-point replies to the comments below.

Reviewer 1

In the present study, the authors reported the results from a systematic cardiovascular screening performed in athletes who recovered form SarS-CoV2 infection. This is timely and interesting study, addressing the concerns raised on myocardial involvement due to COVID-19 and the subsequent potential consequences on professional athletes. The study design is appropriate, all patients underwent a comprehensive evaluation; of note, all of them recieved a similar evaluation (albeit with CMR just in a subgroup) prior to SarS-CoV2 infection. The manuscript is well written and results clearly reported. Out of 47 patients, only 2 were found having abnormal CMR findings (pleural effusion in one case, myocarditis in the other); of note, both patients had newly detected abnormality in other non-imaging test. The authors concluded highlighting the importance of of aCV evaluation priori to RTP in athletes after  SarS-CoV2 infection, albeit suggesting a role for CMR just as a second-line, non-screening, test. I have just few minor comments:

- I cannot see the cited "supplementary data" accompanying the manuscript

We would thank the reviewer for the comment. We uploaded again the supplementary data. We hope that are now available.

- I might be worth adding few details: were all patients evaluated in the pre-vaccination period? Was there any peculiar course of SarS-CoV2 infection in the 3 patients with newly detected CV abnormalities?

We would thank the reviewer for raising these points. We added in the text that all athletes were evaluated in the pre-vaccination period.  Moreover we added the COVID-19 clinical course for each of the three athletes with newly detected CV abnormalities.

- Few typos (i.e. ap-pears, map-ping etc.)

We would thank the reviewer for noticing the errors. We checked the manuscript which was corrected.

Reviewer 2 Report

Dear Viviana Maestrini,

The manuscript you submitted 'Systematic cardiovascular screening in Olympic athletes before and after SARS-CoV-2 infection' aimed to evaluate the prevalence of cardiac involvement after COVID-19 in Olympic athletes. 

Although competitive athletes are usually young and healthy and may develop SARS-CoV-2 infection asymptomatically or with mild symptoms concerns exist about the potential Covid-19 cardiac complications among athletes and the risk of cardiac involvement leading to sport-related arrhythmias and malignant events. 

For this reason, cardiovascular involvement after SARS-CoV-2 infection in patients without symptoms or mildly symptomatic generated concerns on the safety of the return-to-play (RTP) in the athletic population. 

Your study showed the efficacy of the clinical assessment including exercise-ECG to raise suspicion for cardiovascular abnormalities after COVID-19 and demonstrated that the routine use of CMR as a screening tool appears not justified.

The study is well conducted, and the methods include all the screening cardiovascular methods to evaluate cardiac involvements after COVID-19 infection.

The references should improve with recent data:

-       Myocarditis in Athletes Recovering from COVID-19: A Systematic Review and Meta-Analysis. Modica G et al. Int J Environ Res Public Health. 2022 Apr 2;19(7):4279.

-       SARS-CoV-2 infection and return to play in junior competitive athletes: is systematic cardiac screening needed? Cavigli L et al Br J Sports Med. 2022 Mar;56(5):264-270

The results supported the hypothesis that only a minority of athletes had newly detected cardiac abnormalities requiring a temporary withdrawn from competition.

These data confirm that cardiac complication could occur after COVID-19 with a low prevalence and underlying that the routine use of CMR as a screening tool appears not justified.

Author Response

Reviewer 2

The manuscript you submitted 'Systematic cardiovascular screening in Olympic athletes before and after SARS-CoV-2 infection' aimed to evaluate the prevalence of cardiac involvement after COVID-19 in Olympic athletes. Although competitive athletes are usually young and healthy and may develop SARS-CoV-2 infection asymptomatically or with mild symptoms concerns exist about the potential Covid-19 cardiac complications among athletes and the risk of cardiac involvement leading to sport-related arrhythmias and malignant events. For this reason, cardiovascular involvement after SARS-CoV-2 infection in patients without symptoms or mildly symptomatic generated concerns on the safety of the return-to-play (RTP) in the athletic population. 
Your study showed the efficacy of the clinical assessment including exercise-ECG to raise suspicion for cardiovascular abnormalities after COVID-19 and demonstrated that the routine use of CMR as a screening tool appears not justified. The study is well conducted, and the methods include all the screening cardiovascular methods to evaluate cardiac involvements after COVID-19 infection. The references should improve with recent data:

-       Myocarditis in Athletes Recovering from COVID-19: A Systematic Review and Meta-Analysis. Modica G et al. Int J Environ Res Public Health. 2022 Apr 2;19(7):4279.

-       SARS-CoV-2 infection and return to play in junior competitive athletes: is systematic cardiac screening needed? Cavigli L et al Br J Sports Med. 2022 Mar;56(5):264-270 

The results supported the hypothesis that only a minority of athletes had newly detected cardiac abnormalities requiring a temporary withdrawn from competition. 

These data confirm that cardiac complication could occur after COVID-19 with a low prevalence and underlying that the routine use of CMR as a screening tool appears not justified.

We would thank the author for the comment. We agree that these two interesting papers would improve the quality of the manuscript. We updated the references and we added the following sentences:

“A recent metanalysis estimated a pooled prevalence of COVID-19-related myocarditis among athletes ranging from 1 to 4% [25].”

“Very low prevalence of uncommon ventricular arrhythmias was confirmed also in the pediatric population recovering from SARS-CoV-2 infection [28].”

We also checked the manuscript for misspellings.